# Systematic Review of Zinc’s Benefits and Biological Effects on Oral Health

**DOI:** 10.3390/ma17040800

**Published:** 2024-02-07

**Authors:** Silvia Caruso, Chiara Valenti, Lorella Marinucci, Francesca Di Pasquale, Claudia Truppa, Giulia Di Benedetto, Sara Caruso, Stefano Pagano

**Affiliations:** 1Department of Life, Health and Environmental Sciences, Paediatric Dentistry, University of L’Aquila, 67100 L’Aquila, Italy; silvia.caruso@univaq.it (S.C.); giuliadib90@gmail.com (G.D.B.); saracaruso2704@gmail.com (S.C.); 2CISAS “Giuseppe Colombo”, University of Padua, Via Venezia, 15, 35131 Padua, Italy; chiara94.valenti@gmail.com; 3Department of Medicine and Surgery, Faculty of Dentistry, University of Perugia, S. Andrea Delle Fratte, 06156 Perugia, Italy; francesca.dipasquale@outlook.it (F.D.P.); claudiatheodoratruppa@gmail.com (C.T.); 4Department of Medicine and Surgery, Section of Biosciences and Medical Embryology, University of Perugia, 06132 Perugia, Italy; lorella.marinucci@unipg.it

**Keywords:** zinc, dental materials, bio-mechanical properties, oral health

## Abstract

Background: This review was based on the following question: “What is the state-of-the-art regarding the effect of zinc exposure in the oral cavity on a population of adults and children, compared to dental products containing materials other than zinc, considering in vivo (clinical trials and observational studies) and in vitro studies?” according to a PICOS strategy format. This study aims to analyze zinc application in dental materials, with different compositions and chemical formulations, considering how mechanical and biological properties may influence its clinical applicability. Methods: In vivo (clinical trials: controlled clinical trials (CCTs) and randomized controlled trials (RCTs); and observational studies: case control and cohort studies) trials or in vitro studies published in English or Italian during the last 10 years on children and adult patients with zinc exposure were included by three different reviewers using the MEDLINE (via PubMed), Scopus, and Web of Science electronic databases. Results: Titles and abstracts were evaluated following the eligibility criteria. The full texts of eligible studies were then reviewed against the inclusion/exclusion criteria. Scientific and technical information of the 33 included studies were collected into evidence tables, reporting data on in vivo and in vitro studies. A narrative approach was adopted. Conclusions: Antibacterial activity was found to be the most studied property of zinc, but further investigations are needed to establish adjuvant zinc therapies in patients with oral disease.

## 1. Introduction

Zinc (Zn) is a very important trace element, and maintaining its optimal level in the human body is very important for healthy growth and development [1,2]. In the human body, zinc is found in muscles (60%), bones (30%), and skin (5%) [3]. Its functions include its involvement in the activation of various enzymes and proteins [4]; it also contributes to the absorption of vitamin A, vitamin E, and folate [2]. Low zinc levels may be associated with an increased chance of developing infections and degenerative diseases [2]. As with other micronutrients that fall into the mineral category, the amount of zinc needed varies from 2 to 13 mg/day depending on the life stage and sex of the individual, with the upper limit of zinc set at 40 mg/day [5,6,7].

In the oral cavity, it is naturally present at various sites such as saliva, dental plaque, and dental hard tissues such as the hydroxyapatite of dental enamel [5]. The concentration of zinc in the enamel surface varies between 430 and 2100 parts per million (ppm), and its deposition occurs mainly before tooth eruption [6]. Zinc is also important for maintaining periodontal health because of its local and immunological effect on the soft tissues of the oral cavity. It contributes to the formation of healthy teeth [5] and is used in mouthwashes and toothpastes for its important role in preventing plaque and tartar formation [3] and reducing bad breath [3]. 

Zinc has been involved in the composition of dental biomaterials and orthodontic materials due to its immune properties and its effects on cell division and skeletal development [8]. Clinical studies have also shown that zinc ions reduce the rate of enamel demineralization [3]. It has been widely demonstrated to be effective against common prevalent oral health problems, such as caries, gingivitis, and periodontitis. Due to its antibacterial and anti-matrix metalloproteinase activity, zinc has been incorporated into experimental composite resins [9], glass ionomer cements [10], adhesive resins [11,12], and desensitizing solutions [13], the latter being particularly useful for the treatment of root dentin demineralization [14].

In recent decades, many researchers have also reported the antibacterial activity of zinc oxide nanoparticles (ZnONP) [15] on various microorganisms, such as *Escherichia coli* [16], *Campylobacter jejuni* [17], *Pseudomonas aeruginosa* [18], and *Vibrio cholerae* [19]. Currently, modern nanotechnology is moving toward the study of zinc-based materials to produce, for example, light-curing restorative composite resins and related adhesive systems, impression materials, dental implants with surfaces coated in zinc, mouthwashes, and root canal fillings with an antibacterial effect [20,21]. 

However, although positive effects of new zinc-containing dental materials have been shown in the literature, in order to completely replace the traditionally used materials, in vivo studies would be needed to confirm the properties under real conditions and with a larger sample of patients. In addition, the potential toxicity of nanomaterials has attracted the attention of researchers, but there are no recent studies on this issue [22,23].

The purpose of this review is to investigate the state of the art with respect to the application of zinc in dental materials in different dental fields and with different compositions and chemical formulations, considering how mechanical and biological properties may influence its clinical applicability.

Null hypothesis: the addition of zinc to dental materials affected biological and mechanical properties.

## 2. Materials and Methods

This review was conducted based on the following question: “What is the state-of-the-art regarding the effect of zinc exposure in the oral cavity on a population of adults and children, compared to dental products containing materials other than zinc, considering in vivo (clinical trials and observational studies) and in vitro studies?” This question was created according to the PICOS strategy format [24] (Table 1) and was not registered online.

The present study will include studies published in English or Italian published in the last 10 years, including in vivo trials (clinical trials: controlled clinical trials (CCTs) and randomized controlled trials (RCTs); and observational studies: case control and cohort studies) and in vitro studies on children and adult patients with zinc exposure as a variable and no zinc exposure as comparator/control. 

Exclusion criteria: systematic reviews, animal studies, case reports, review articles, editorials, opinions, surveys, guidelines, conferences, commentary articles, studies on other chemical compounds, studies with no full-text available.

Primary outcome: Oral health.

Secondary outcome: None.

The search process was performed by three different reviewers using the MEDLINE (via PubMed), Scopus, and Web of Science electronic databases. The search strategy was outlined based on PubMed MeSH terms, as shown in Table 2, and adapted for the other databases.

This review was conducted based on the PRISMA statement and checklist [25] (Appendix A, available online). The screening process was performed on the titles, abstracts, and full-texts of the articles that potentially met the inclusion criteria. In each phase, three reviewers independently assessed each article. Any disagreement was resolved through discussion or by consulting other authors. The PRISMA flow diagram (Figure 1) was used to report the included and excluded studies. The electronic search was followed by a manual search of the references list of included articles.

Three independent reviewers evaluated the titles of the articles initially retrieved in the search following the eligibility criteria, and those with no relevance were excluded. Articles compatible with the inclusion criteria were selected for further examination and the abstracts were screened. The full texts of potentially eligible studies were then independently reviewed against the inclusion/exclusion criteria by the reviewers, and any disagreement was resolved by consultation with the other authors. Scientific and technical information were collected into two evidence tables (Table 3 and Table 4) with Microsoft Office Excel, reporting data about in vivo and in vitro studies, respectively, with Table 3 including the following: author(s) and year of publication, population characteristics, zinc application and chemical composition, control group, biological properties and effects on oral health, principal findings, fundings, and quality assessment score; and Table 4 including the following: author(s) and year of publication, fundings, cell types or microbial strains, zinc application and chemical composition, specimen characteristics, control group, biological properties and effects on oral health, principal findings, and quality assessment score. For data analysis, a narrative approach was adopted.

The Quality Assessment for Diverse Studies (QuADs) tool [26] (Appendix A, available online) was selected and performed by two independent reviewers for each included study to assess the inherent quality, assigning a reliability score. The tool consists of a set of 13 items with a score from 0 (incomplete information) to 3 points (complete information).

## 3. Results

The flow diagram (Figure 1) shows a total of 442 papers after the electronic search, with 392 potentially eligible articles in English. A total of 261 studies were excluded after title screening. Then, abstract screening was completed on 131 studies, and 62 papers were eligible for full-text review. Finally, 33 studies were included for data extraction. Two articles were published in 2017 [27,28], ten in 2018 [12,13,29,30,31,32,33,34,35,36], four in 2019 [37,38,39,40], seven in 2020 [41,42,43,44,45,46,47], nine in 2021 [48,49,50,51,52,53,54,55,56], and one in 2022 [57].

According to the quality assessment tool, 33 studies met the criteria and were considered reliable. The highest scores were 38/39 points in four studies [32,36,40,53] and 36/36 in three studies [13,48,55], while the lowest scores were 32/36 [27,28,47,51,52] and 34/39 [34,41] (Table 3 and Table 4, and Appendix A, available online).

Considering the study design, five studies were in vivo [27,29,37,41,48] and 28 studies were in vitro [12,13,28,30,31,32,33,34,35,36,38,39,40,42,43,44,45,46,47,49,50,51,52,53,54,55,56,57].

Of the in vitro studies, the following 16 papers on the effects of zinc were on human cell populations: gastrointestinal apparatus using liver cancer cell line [49]; gingival fibroblasts were analyzed in the orofacial district [36,40,53,54,55,56]; oral keratinocyte cells [44]; pulp fibroblasts [12] or dental pulp cells [28,45,47]; and epithelial tissue and saliva-derived biofilm models [31]. For the urinary system, the embryonic kidney cell line was used [49]; and for the hematopoietic system, red blood corpuscles [47], bone marrow mesenchymal stem cells [51], and umbilical vein endothelial cells [56] were applied. For the integumentary apparatus, fibroblast cells were used instead [32]; and for skeletal system, the osteosarcoma cell line [52] was used.

Eleven studies used different *S. mutans* microbial strains [12,30,32,33,36,38,39,42,43,45,57], and four studies used *E. faecalis* [32,34,39,46]. In four studies, *P. gingivalis* was considered [33,35,46,51], and in three studies, *F. nucleatum* [33,35,46]. *A. actinomycetemconcomitans* was used in two studies [35,46] and *P. intermedia* in two studies [33,46]. Two studies investigated *Staphylococcus aureus* [46,47], while only one study used *Streptococcus aureus* [51]. Other cariogenic microorganisms analyzed were *S. oralis* [35]; *S. sobrinus, S. salivarius*, and *S. anginosus* [33]; *L. paracasei* [46]; *L. fermentum* [39]; *E. coli* [52]; and *A. naeslundii* and *V. parvula* [35]. *C. albicans* was used in two studies [39,46]. Vergara-Llanos et al. studied mono and multispecies bacterial models [53].

### 3.1. In Vivo Studies 

As reported in Table 3, zinc mechanical properties, like restorations adhesive strength or flexural strength, were not considered in any of the five included studies. Regarding the biological properties evaluated, three studies analyzed the microbiological properties and effects of many bacterial species of different plaque samples using toothpaste formulations with zinc [29,37,41]; and two studies evaluated antibacterial effects considering endodontic tissue repair and lesion sterilization after treatments with zinc oxide eugenol paste (ZOE) [48] and zinc oxide-ozonated oil [27]. The clinical and radiographic effectiveness of pulpectomy agents were also taken into consideration [27,48]. 

Prasad et al. [29] observed that toothpastes with 0.96% zinc (zinc oxide, zinc citrate), 1.5% L-arginine, and either 1450 ppm or 1000 ppm of fluoride caused significant reductions in oral bacteria compared to fluoride alone. Sreenivasan et al. [41] confirmed these findings, while Hagenfeld et al. reported no significant difference in zinc-substituted carbonated hydroxyapatite dentifrice compared to toothpaste containing an amine fluoride/stannous fluoride [37].

Doneria et al. highlighted how zinc oxide-ozonated oil had 100% endodontic clinical success and was comparable to Vitapex formulation at 6 and 12 months [27]. Moura et al. also reported the same clinical and radiological efficacy of zinc oxide eugenol (ZOE) paste compared to a formulation of chloramphenicol, tetracycline, and zinc oxide [48].

### 3.2. In Vitro Studies

Information about the in vitro studies is reported in Table 4, considering applications in various dental fields: conservative and cariology, endodontics, periodontology and implantology, orthodontics, and other aspects.

**Table 3 materials-17-00800-t003:** Characteristics of in vivo studies included.

Author(s) and Year of Publication	Population Characteristics	Zinc Application and Chemical Composition	Control Group	Zinc Biological Properties and Effects on Oral Health	Principal Findings	Fundings	Quality Assessment Score
Doneria et al. 2017 [27]	N = 64 primary molars of 43 children (aged between 4 and 8 years)	Zinc oxide–ozonated oil (ZnO-OO): zinc oxide powder (DPI) (0.2 g, arsenic free) and ozonated castor oil (0.007 cc Ozonil, Ozone Forum of India) using motor-driven lentulospirals	Vitapex (Neo Dental Co.) and modified 3Mix MP paste	Clinical and radiographic success, effectiveness as pulpectomy agents: follow up at 1, 6, and 12 months (RX at 6 and 12 months).	Clinical success of ZnO-OO and Vitapex were comparable at 6 and 12 months (100%). Radiologically, success rates for ZnO-OO were 100% at 6 and 12 months, with significant differences between three groups after 12 months (*p* = 0.029 and *p* = 0.011).	None	32/36
Hagenfeld et al. 2019 [37]	N = 41 nonsmokers patient (mean age 54.86 ± 10.19 years; 25 females and 16 males) with PPD ≥ 4 mm in a minimum of 4 teeth, aged ≥ 18–75 years, and with at least 10 natural teeth	Zinc-substituted carbonated hydroxyapatite dentifrice: HA group N = 20 (BioRepair, Wolff)	dentifrice containing an amine fluoride/stannous fluoride: AmF/SnF_2_ group N = 21 (Meridol, CP GABA)	Microbiome variation analysis with paired-end Illumina Miseq 16S rDNA sequencing: plaque samples from buccal/lingual, interproximal, and subgingival sites at baseline, 4 weeks after oral hygiene, and 8 weeks after periodontal therapy.	No significant difference observed in terms of changes on community level (alpha and beta diversity) and on the level of single agglomerated ribosomal sequence variants (aRSV). In interproximal and subgingival sites: Fusobacterium and Prevotella species associated with periodontitis.	Kurt Wolff GmbH	35/39
Moura et al. 2021 [48]	N = 88 primary molars with pulp necrosis (mean age 5.5 ± 1.2 SD, 35 males and 35 females)	Zinc oxide eugenol paste (ZOE) (N = 44), with the zinc oxide packed into 250 mg capsules and mixed with 0.1 mL eugenol (Biodynamics); chemical–mechanical canal debridement and disinfection with 2% chlorhexidine solution (LT Rioquímica) and K-files (sizes 15 to 25; Dentsply); restoration with high-viscosity glass ionomer cement (Gold Label 9R, GC)	CTZ group (N = 44): 62.5 mg of chloramphenicol, 62.5 mg of tetracycline, and 125 mg of zinc oxide	Effectiveness of lesion sterilization and tissue repair (LSTR), evaluation every 3 months for 12 months, and clinical and radiographic evaluation.	No significant difference observed. The mean time taken to perform was 145.1 (±53.2) minutes for ZOE (*p* < 0.001). At 12 months, the clinical success rate was 90.9% and the radiographic success rate 72.7% for ZOE.	N/A	36/36
Prasad et al. 2018 [29]	173 subjects	New fluoride toothpastes with Dual Zinc plus Arginine formulations: zinc (zinc oxide, zinc citrate) 0.96%, 1.5% Arginine, and 1450 ppm fluoride; zinc (zinc oxide, zinc citrate) 0.96%, 1.5% Arginine, and 1000 ppm fluoride (Dual Zinc plus Arginine; Colgate-Palmolive Company)	Regular fluoride dentifrice containing 1450 ppm fluoride (Colgate dentifrice; Colgate-Palmolive Company)	Effect on reducing bacteria in oral biofilm (CFU): oral samples collected from teeth, tongue, oral buccal mucosa, gingiva, mand saliva at baseline and 12 h after 14- and 29-days of assigned product use.	Subjects using the Dual Zinc plus Arginine Toothpaste with 1450 ppm F exhibited significant reductions in bacteria on buccal (35.4%, *p* < 0.001), teeth (38.3%, *p* < 0.001), gingiva (25.9%, *p* = 0.043), tongue (39.7, *p* = 0.001), and in saliva (41.1%, *p* < 0.001) 12 h after 29 days of product use. Toothpastes containing 0.96% zinc (zinc oxide, zinc citrate), 1.5% L-arginine, and either 1450 ppm or 1000 ppm fluoride as sodium fluoride in a silica base provide significant reductions in oral bacteria compared to toothpaste with fluoride alone.	Colgate-Palmolive Company	37/39
Sreenivasan et al. 2020 [41]	N = 44 adults (19-63 years) with at least 20 natural teeth, in good general health, and with a plaque index scores of 1.5 or more and gingival index scores of 1.0 or more	Herbal toothpaste incorporating zinc (N = 22, mean age 46.2)	Commercially available fluoride toothpaste (Colgate Dental Cream, Great Regular Flavor) (N = 22)	Microbiologic analysis (CFU) on anaerobic organisms, Gram-negative bacteria and malodor bacteria of dental plaque, tongue scrapings, and cheek surfaces.	Significant reductions in functional bacterial groups from distinct oral niches compared to control group (*p* < 0.05). Reductions between 42 and 68% for anaerobic bacteria 12 h after brushing, increasing to 46–80% 4 h after brushing; and between 49 and 61% for Gram-negative bacteria, that increased to 54–69% 4 h post-brushing.	Colgate-Palmolive Company	34/39

**Table 4 materials-17-00800-t004:** Characteristics of in vitro studies included.

Author(s) and Year of Publication	Fundings	Cell Types or Microbial Strains	Zinc Application and Chemical Composition: Specimen Characteristics	Control Group	Zinc Biological Properties and Effects on Oral Health	Principal Findings	Quality Assessment Score
**Conservative and Cariology**							
Barma et al. 2021 [49]	None	Human liver cancer (Hep G2) and human embryonic kidney 293 (HEK-293T) cell lines	Zinc oxide nanoparticles synthetized (ZnO-NP) varnish	None	Inhibition of *S. mutans* growth, biofilm, acid production, and antioxidant potential with 2,2-diphenyl-2-picrylhydrazyl hydrate (DDPH) assay, and cytotoxicity; ZnO-NP characterized using UV spectroscopy, x-ray diffraction spectroscopy, and transmission electron microscopy; secondary metabolites assessed using fourier transform infrared spectroscopy.	Excellent antimicrobial properties against *S. mutans*; minimum inhibitory and bactericidal concentrations were 0.53 μg/mL and 1.3 μg/mL, respectively. A total of 0.1 mg/μL had the greatest zone of inhibition (24 mm). A total of 0.1 mg/μL inhibited 90% of *S. mutans* biofilms and exhibited antioxidant capacity in a dose-dependent manner (94% inhibition, 100 μg/mL). A total of 0.1 mg/μL ZnO-NP caused very low cytotoxicity to Hep G2 cells and was non-cytotoxic to HEK-293T cells.	35/36
Eskandarizadeh et al. 2019 [38]	None	*S. mutans* strain	ZDMA powder: 100 mL of Hexan, 0.03 mL of Triton100,and 8.4 gr of ZnO; 5 test groups with dental resin adhesive containing zinc dimethacrylate ionomer (ZDMA) in different concentrations into resin bonding: 0.5, 1, 2.5, and 5 wt.%	Pure resin adhesive (Tetric N-Bond, voclar Vivodent)	Antibacterial test (CFU): bacterial strains and growth conditions (*S. mutans* PTCC 1683, Persian Type Culture Collection, IROST) for direct contact and material aging; physical test: degree of conversion (DC) and Zinc ion release amount in aqueous medium; mechanical test: compressive strength and shear bond test (enamel and dentine separately).	Significantly reduced amount of S. Mutans (*p* < 0.05); DC was enhanced; ion release analysis revealed stability of Zn^2+^ (as in the 5 wt.% group); even after 9 cycles of a 24 h wash. Compressive strength was significantly reduced (*p* < 0.05) just in the 5% ZDMA group, while the other groups were superior in comparison to the control. For the dentine shear bond strength, only the 5% ZDMA group was significantly higher than the control (*p* = 0.000).	33/36
Garcia et al. 2021 [50]	None	None	ZnO incorporated at 2.5 (G_2.5%_), 5 (G_5%_), and 7.5 (G_5%_) wt.% in an experimental dental adhesive	ZnO incorporated at 0 (G_CTRL_) in dental adhesive	Chemical and mechanical properties: degree of conversion (DC), flexural strength (FS), and elastic modulus (E). Antibacterial response in 48 h microcosm biofilm model after the formation of acquired pellicle on samples’ surfaces: colony-forming units (CFU), metabolic activity, and live/dead staining.	DC ranged from 62.21 (±1.05)% for G_Ctrl_ to 46.15 (±1.23)% for G_7.5%_; G_7.5%_ showed lower FS; G_2.5%_ showed higher E; G_7.5%_ had lower CFU/mL compared to G_Ctrl_ for *S. mutans* and total microorganisms, despite presenting lower metabolic activity and higher dead bacteria (*p* < 0.05).	34/36
Huang et al. 2020 [42]	National Natural Science Foundation of China (21371139) and the Graduate Innovation Foundation of Wuhan Institute of Technology (CX2017125)	*S. mutans* (Ingbritt)	Ag/ZnO nanocomposite	None	Effects at sub-minimum inhibitory concentrations (sub-MICs) on virulence factors of *S. mutans* and related genes expressions by growth curves and MTT staining method; biofilm formation with crystal violet staining method and scanning electron microscopy; adherence, cell-surface hydrophobicity, acidogenicity, and extracellular polysaccharides (EPS) of *S. mutans*. Virulence factors related genes expressions by qRT-PCR.	Decrease of 69.00% biofilm formation, 31.78% sucrose-independent and 48.08% sucrose-dependent adherence, 69.44% cell-surface hydrophobicity, and 72.45% water-soluble and 90.60% water-insoluble EPS with Ag/ZnO at 1/2 MIC. Expression of virulence factors-related genes were significantly suppressed.	36/39
Lavaee et al. 2018 [30]	Vice-Chancellory of Shiraz University of Medical Science (Grant No. 8794121)	Standard strain of *S. mutans* (ATCC 35668, PTCC 1683)	Different concentrations of zinc sulfate (Merck) and zinc acetate (Falcon) solutions were prepared in concentrations of 6.25, 12.5, 25, and 50 μg/mL	Penicillin and chlorhexidine	Inhibitory and bactericidal effects on *S. mutans*: diameters of zone of inhibition detected by the disc diffusion method; minimum inhibitory concentration (MIC) and minimum bactericidal concentration (MBC).	Zinc sulfate and zinc acetate salts with 37.19 and 31.25 µgr/mL concentration had an inhibitory effect on *S. mutans*, respectively. MIC and MBC of zinc sulfate solution were higher than penicillin and chlorhexidine (*p* < 0.001), but no priority in antibacterial activity of the studied zinc salts was determined.	37/39
Manus et al. 2018 [31]	N/A	In vitro oral epithelial tissue and saliva-derived biofilm models	Zinc citrate dentifrice formulations prepared with increasing replacement of zinc citrate with zinc oxide (a water insoluble source of zinc ions) to generate a dual-zinc active system	None	Bioavailability enhancement of zinc, zinc delivery, and antibacterial efficacy; zinc penetration and retention with imaging mass spectrometry (I-MS)	Enhanced antibacterial performance observed through significant reductions in metabolic activity as measured through bacterial glycolytic function (*p* = 0.0001) and total oxygen consumption (*p* = 0.0001).	35/36
Mirhosseini et al. 2019 [39]	None	Dtrains of *S. mutans* (ATCC 35668), *E. faecalis* (ATCC 29212), *L. fermentum* (ATCC 14931), and *C. albicans* (ATCC 10231)	Solutions at the concentration of 10 μg/mL were prepared using 20-nm, 40-nm, and 140-nm nano ZnO (nZnO) powder (Research Nanomaterials Inc.)	None	Antimicrobial effect of various sizes and concentrations of zinc oxide (ZnO) nanoparticles on *S. mutans*, *E. faecalis*, *L. fermentum*, and *C. albicans* with a spectrophotometer (UV-150-02; Shimadzu C0) for microbial growth, disk diffusion method, and minimum inhibitory concentrations (MICs), and minimum bactericidal concentrations (MBCs) using broth microdilution method.	The antimicrobial activity of nZnO increases with the decreasing particle size against *S. mutans* (*p* = 0.00), *L. fermentum*, and *E. faecalis* (*p* < 0.02). The greatest antimicrobial effect was observed against *S. mutans* and *E. faecalis*.	33/36
Peralta et al. 2018 [32]	CAPES (Coordination for the Improvement of Higher Education Personnel) and CNPq (National Council for Scientific and Technological Development (Grant No. 313294/2014-3)	*S. mutans* UA159, *Enterococcus faecalis* ATTC4083; fibroblast (NCTC clone 929) cells	Elastomeric temporary resin-based filling materials (TFMs) containing zinc methacrylate (ZM, Aldrich Chemical Co.), with concentration of 0.5% (Z0.5); 1% (Z1), 2% (Z2), or 5% (ZM5) wt% was added to the TFMs; N = 10 disk-shaped (6.0 mm in diameter and 1 mm thick) specimens per group for water sorption and solubility; N = 10 dumbbell-shaped specimens (10 × 5 × 1 mm) for UTS; N = 4 specimens 15 × 6 mm for hardness; N = 8 specimens 6 × 1 mm for biofilm accumulation test	ZM concentration of 0% control and the Fermit-N (Ivoclar Vivadent) (F) used as a commercial reference	Physical and mechanical properties, antibacterial effect, and biocompatibility: microleakage, water sorption/solubility, degree of conversion with Fourier transform infrared spectroscopy, depth of cure with scraping method, ultimate tensile strength (UTS), and the Shore D scale hardness tester were determined and compared. A modified direct contact test (DCT) with *E. faecalis* and a *S. mutans*’ biofilm accumulation assay were carried out to evaluate the antimicrobial effect and cytotoxicity (MTT).	Physical, mechanical, and biological properties are comparable with the properties of the commercial reference. Some properties were improved: lower microleakage and water sorption, and higher ultimate tensile strength values (*p* < 0.001). After the 24 h in direct contact test, all TFMs with ZM were similar (*p* = 0.058), also considering DC (*p* < 0.034).	38/39
Steiger et al. 2020 [43]	None	*S. mutans* (ATCC 25175) and *S. mutans* clinical isolate	Divalent cation Zn^2+^ and in combination ZnCl 1, 3, 10, 30, 100, and 200 mM for antibacterial properties, 1 mM ZnCl_2_ for biofilm analysis, 1, 10, and 100 mM for HA dissolution; hydroxyapatite (HA) disks (5 mm, HiMed Inc.) on biofilm formation	No divalent ions	Biofilm formations and growth using confocal laser scanning microscopy (CLSM) with Leica SP8 microscope (Leica SP8), cariogenic dissolution of hydroxyapatite, EPS extraction of *S. mutans* cultures with the phenol–sulphuric acid assay, analysis of the binding of ZnCl_2_ to EPS via isothermal titration calorimetry (ITC).	No significant effects observed. Zinc inhibited bacterial adhesion was also at low concentrations and had a strong antibacterial effect on the strains as well as on calcium dissolution with less biofilm and less EPS. Zn^2+^ had the lowest affinity to all EPS; the unbound zinc could also still remain in the environment and keep its antimicrobial properties.	34/36
Suzuki et al. 2018 [33]	Supported in part by Grants in Aid of Scientific Research (Nos. 26463175, 15K14423, and 16K07205) from the Ministry of Education, Culture, Sports, Science, and Technology of Japan, from the Sato Fund (2015–2016) of the Nihon University School of Dentistry	*P. gingivalis* FDC 381, *P. gingivalis* W83, *P. gingivalis* ATCC 33277, *F. nucleatum* ATCC 25586, *P. intermedia* ATCC 25611, *S. mutans* JCM 5705, *S. sobrinus* JCM 5176, *S. salivarius* GTC 0215, and *S. anginosus* FW73	metal chlorides, ZnCl_2_, and metal acetates, (CH_3_COO)_2_Zn	Other metal ions	Binding of zinc ions to gaseous hydrogen sulfide (H_2_S); minimum concentration needed to inhibit H_2_S volatilization determined with serial dilution methods; six oral bacterial strains related to volatile sulfur compound (VSC) production and three strains not related to VSC were evaluated; inhibitory effects on growth of oral bacteria.	Zinc ions’ effect on the growth of oral bacteria was strain-dependent. *F. nucleatum* was the most sensitive and suppressed by media containing 0.001% zinc ions. There was an inhibitory effect on oral malodor with direct binding with gaseous H_2_S and suppressing the growth of VSC-producing oral bacteria.	35/39
Tabatabaei et al. 2019 [40]	N/A	Human gingival fibroblasts (HGFs)	16 toothpastes and 4 mouthwashes available in Iranian market containing sodium fluoride (NaF), sodium lauryl sulfate, cocamidopropyl betaine, zinc lactate, paraben, and sodium benzoate	None	MTT assay was used to assess cytotoxicity.	Fifference in cytotoxicity was statistically significant (*p* < 0.001). Cytotoxicity was time- and concentration-dependent. Cytotoxicity of all concentrations of zinc lactate was <50%.	38/39
Toledano-Osorio et al. 2018 [13]	Supported by the Ministry of Economy and Competitiveness (MINECO) and the European Regional Development Fund (FEDER) (Project MAT2017-85999-P)	N = 30 and N = 15 sound single-rooted teeth obtained with informed consent from donors (18 to 25 yr of age); each root was removed 5 mm below the cement-enamel junction using a low-speed diamond saw (Accutom-50 Struers); N = 15 sound single-rooted teeth; from each tooth, two dentin blocks were obtained from the buccal surface of the root, just below the cementodentinal junction; the tooth was cut perpendicular to the axial axis using a diamond saw (Accutom-50 Struers)	Zinc-doped NPs (Zn-NPs), PolymP-n active nanoparticles (NanoMyP): aqueous solutions of ZnCl_2_ (containing zinc at 40 ppm at pH 6.5)	None	Dentin hypersensitivity: field emission scanning electron microscopy (FESEM Gemini, Carl Zeiss), energy dispersive analysis using an X-ray detector system (EDX Inca 300, Oxford Instruments), atomic force microscope (AFM Nanoscope V, Digital Instruments, Veeco Metrology group) and Nano-DMA analysis; complex, storage, loss modulus, and tan delta (δ).	Treating dentin with Zn-nanoparticles, complex modulus values attained at intertubular and peritubular dentin were higher.	36/36
Xu et al. 2020 [44]	Financial support from the National Natural Science Foundation of China (Grant Nos. 51973133, 51925304, 51773128, and 21534008)	Human oral keratinocyte (HOK) cells	Zinc-substituted hydroxyapatite/alendronate-grafted polyacrylic acid hybrid nanoneedles (ZHA@ALN-PAA) 286 mg of Zn(CH_3_COO)_2_%2H_2_O	None	Cytotoxicity ws assessed via Cell Counting Kit-8 (CCK-8 assay).	Cell viability higher than 80%. The acceptable cytocompatibility of the nanomaterials guarantees their further applications.	36/39
**Endodontics**							
Fan et al. 2018 [34]	Financially supported by the National Natural Science Foundation of China (Grant Nos. 81570969 and 81470732)	*E. faecalis* (ATCC 29212, ATCC), human extracted wisdom teeth	A serial of Ag^+^-Zn^2+^ atomic combination ratios (1:1, 1:3, 1:6, 1:9, or 1:12) tested on both planktonic and biofilm-resident *E. faecalis*: 1 mL suspension (1 × 103 CFUs/mL) for CFU for 24 h, 4 mL suspension (1 × 103 CFUs/mL) for dynamic antimicrobial effect at 1, 2, 3, 4, and 5 h (2, 4, 6, 8, and 10 h for the Zn^2+^ only group); dentin slices (4 × 4 × 1 mm) prepared from human teeth for biofilm formation	CHX	Co-work pattern and optimum ratio between Ag^+^ and Zn^2+^ synergy, antibacterial activity against *E. faecalis* (CFU), dynamic growth curve method using spectrometer, serial microdilution assay for MIC and MBC, inhibition of biofilm on dentin, using a Zn^2+^ pretreatment study, membrane potential-permeability measurement (BacLight bacterial membrane potential kit (Molecular Probes, Invitrogen)) and flow cytometry; cytotoxicity of various Ag^+^-Zn^2+^ atomic ratios with cell counting kit-8 (CCK-8) (Dojindo Laboratories).	Ag^+^-Zn^2+^ (1:9 and 1:12) had the most powerful ability against planktonic and biofilm-resident *E. faecalis* (*p* < 0.05). This co-work could be attributed to the depolarization of *E. faecalis* cell membrane via the addition of Zn^2+^.	34/39
Garcia et al. 2018 [12]	Coordination for the Improvement of Higher Education Personnel (CAPES) for the scholarship of GARCIA IM (n° 1678704)	Pulp fibroblasts and *S. mutans* (NCTC 10449)	Zinc oxide quantum dots (ZnOQDs) into adhesive resin: ZnOQDs synthesized by sol–gel process and incorporated into HEMA; wxperimental adhesive resin formulated mixing 66.6 wt.% BisGMA and 33.3 wt.% HEMA	Photoinitiator system as control group	Antibacterial activity against *S. mutans* with direct contact inhibition evaluation and cytotoxicity with SRB assay.	Antibacterial activity assay indicated a significant difference (*p* = 0.003), with a reduction of more than 50% of biofilm formation on ZnOQDs group.	35/39
Garcia et al. 2020 [45]	Financed in part by the Coordenação de Aperfeiçoamento de Pessoal de Nível Superior—Brasil (CAPES)—Finance Code 001, and Conselho Nacional de Desenvolvimento Científico e Tecnológico—Brasil (CNPq)—n° 307095/2016-9.	Human pulp cells collected from a third molar extracted; *S. mutans* strain (NCTC 10449)	Zinc-based particle with ionic liquid as filler for an experimental adhesive resin: zinc chloride (ZnCl_2_) used to synthesize 1-n-butyl-3-methylimidazolium trichlorozincate (BMI.ZnCl3), hydrolyzed under basic conditions to produce simonkolleite (SKT) particles, incorporated at 1, 2.5, or 5 wt.% in adhesive, containing bisphenol A glycerolate dimethacrylate (Bis-GMA) mixed with 2-hydroxyethyl methacrylate (HEMA) at a proportion of 66:33 wt.%; disc-shaped samples (1 × 4 mm); N = 3 for antibacterial activity; N = 5 for cytotoxicity; N = 5 for DC; N = 5 for hardness; hourglass-shaped samples (8 × 2 × 1 mm) N = 10 for UTS	Group without SKT as control	Scanning electron microscopy and transmission electron microscopy for SKT analysis.; antibacterial activity against *S. mutans* (CFU), cytotoxicity (SRB assay), degree of conversion (DC) using FTIR-ATR, ultimate tensile strength (UTS), Knoop hardness, and softening in solvent.	SKT addition provided antibacterial activity against biofilm formation and planktonic bacteria (*p* < 0.05). No changes in pulp cells’ viability. DC extended to 64.44 (± 1.55)% for 2.5 wt.%, but was not significant, like UTS, and softened in solvent. Physicochemical properties of adhesives were not affected by SKT incorporation.	36/39
Pilownic et al. 2017 [28]	N/A	Dental pulp cells	Zinc oxide eugenol (ZOE), Vitapex, and Calen paste thickened with zinc oxide (ZO); N = 5 for every analysis: 1, 4, and 12 h and 1, 3, 7, 15, and 30 days for ph; 1, 4, 12, and 24 h for direct contact test; 1, 3, and 7 days for MTT assay	Experimental MTA-based material	pH, radiopacity, and antimicrobial effect (direct contact test) against *E. faecalis*, cytotoxicity (MTT assay), and biocompatibility test	No significant effects observed. Vitapex presented the highest cell viability.	32/36
**Periodontology and Implantology**							
Chen et al. 2021 [51]	N/A	rBMSCs; bone marrow mesenchymal stem cells; *S. aureus* and *P. gingivalis*	Co-incorporated zinc- (Zn-) and strontium- (Sr-) nanorod coating on sandblasted and acid-etched (SLA) titanium (SLA-Zn/Sr) fabricated by hydrothermal synthesis	SLAactive titanium	Osteogenesis and inhibition of biofilm formation	Sufficient interface bonding strength (42.00 ± 3.00 MPa). SLA-Zn/Sr enhanced the corrosion resistance property of Ti. SLA-Zn/Sr promoted cellular initial adhesion, proliferation, and osteogenic differentiation while inhibiting the adhesion of *S. aureus* and *P. gingivalis* and down-regulating icaA gene expression, reducing polysaccharide secretion and suppressing *S. aureus* biofilm formation.	32/36
Fröber et al. 2020 [46]	N/A	*Fusobacterium nucleatum*, *Porphyromonas gingivalis*, *Prevotella intermedia*, *Aggregatibacter actinomycetemcomitans*, *Enterococcus faecalis*, *Staphylococcus aureus*, *Lactobacillus paracasei*, and *Candida albicans*	Glucose-1-phosphate (Glc-1P) biofunctionalized zinc peroxide (ZnO_2_) nanoparticles of four different synthesis ratio (1–10:1) and sizes (4–5 nm)	Nanoparticles stabilized with o-phosphorylethanolamine, bis [2-(methacryloyloxy)ethyl] phosphate or dioctyl sulfosuccinate used as controls	Antimicrobial properties: minimal inhibitory (MIC) and minimal microbicidal concentration (MBC or MFC) determined under different pH conditions; transmission electron (TEM) and fluorescence microscopy after live-dead-staining performed on selected combinations of pathogens and nanoparticles to visualize interactions.	Inhibitory effect on Gram-negative anaerobes and on A. actinomycetemcomitans with a pH-dependent MIC ≥ 25 μg/mL and MBC ≥ 50 μg/mL. In TEM images, the attachment of nanoparticle chains to the bacterial outer membrane and the subsequent penetration were found together with an intracellular oxygen release.	33/36
Lin et al. 2021 [52]	N/A	MG-63 cells; human osteosarcoma cell line; *Escherichia coli*	Inner layer of nanoporous TiO_2_ and the outer layer of the chitosan matrix with ZnO nanoparticles	Chitosan coating alone or pure Ti	Dental implant-related infections: antibacterial activity against *E. coli*; the effects of the amount of ZnO coating on wettability, anti-scratch ability, bioactivity, and corrosion resistance.	Improvement in antibacterial properties and bioactivity of the chitosan/ZnO coating attributed to Zn^2+^ ions release. The critical force of scratching was approximately twice that of the chitosan coating. The potentiodynamic polarization confirmed that the corrosion resistance was improved. Good cytocompatibility in MG-63 cells as compared to pure Ti.	32/36
Sánchez et al. 2018 [35]	N/A	Static subgingival biofilm model with *Streptococcus oralis*, *Actinomyces naeslundii*, *Veillonella parvula*, *Fusobacterium nucleatum*, *Porphyromonas gingivalis,* and *Aggregatibacter actinomycetemcomitans*	Polymeric PolymP-n Active nanoparticles; hydroxyapatite discs coated with nanoparticles (NPs) doped with zinc (12, 24, 48, and 72 h)	PBS as control; NPs alone and doped with calcium, silver, and doxycycline	Nano-roughness of the different disc surfaces (SRa, in nm); morphological characteristic of the biofilms (thickness (μm) and bacteria viability) studied with different microscopy modalities; q-PCR to assess the effect of the nanoparticles on the bacterial load CFUmL^−1^.	Surfaces containing nanoparticles showed significant increments in roughness compared to controls (*p* < 0.05). Reductions in bacterial viability was more pronounced with silver and doxycycline NPs.	34/36
Vergara-Llanos et al. 2021 [53]	Viscerrectoría de Investigación y Desarrollo, Universidad de Concepción, Chile, grant nº 216.102.024-1.0, andViscerrectoría de Investigación, Desarrollo y Creación Artística, Universidad Austral de Chile	Human gingival fibroblasts(HGFs) and mono and multispecies bacterial models	Zinc oxide nanoparticles (ZnO-NPs)	Copper nanoparticles (Cu-NPs)	MIC and spectral confocal laser scanning microscopy; cytotoxic effects by MTT, LDH assays, production of ROS, and the activation of caspase-3	After 24 h, ZnO-NPs are biocompatible between 78 and 100 μg/mL. Antibacterial activity in a monospecies model is strain-dependent, and in a multispecies model was in lower doses after 10 min of exposure. With induced mitochondrial dose-dependent cytotoxicity, ZnO-NPs increase LDH release and intracellular ROS generation. In a multispecies model, a significant decrease in the total biomass volume (μ_3_) and bactericidal activity was observed with 125 μg/mL (*p* < 0.05).	38/39
**Orthodontics**							
Kachoei et al. 2021 [54]	Iran National Science Foundation (INSF) (Grant No. 92033574)	HGF (human gingival fibroblast) cells. N = 120 extracted human maxillary first premolars for for four groups of bracket bonding evaluation: composite without nanoparticles (O); with ZnO (Z) nanoparticles; with ZnO nanoparticles and silver ions (A&Z); and with Ag/ZnO nanoparticles (AZ) synthesized using optical precipitation	New bioactive orthodontic composite resin containing silver/zinc oxide (Ag/ZnO) nanoparticles; disc-shaped specimens (5 mm × 1 mm, N = 5) from composite for ion release e valuation	None	Wettability, shear bond strength (SBS) test, Zn and Ag release (inductively coupled plasma optical emission spectrometry (ICP-OEP, 700, Agilent Technologies)), cytotoxicity (MTT assay), biocompatibility, and antimicrobial properties with microbroth dilution (MIC) method against *S. mutans*, Lactobacillus, and *C. albicans*.	Significant antimicrobial properties (*p* < 0.05). Based on the MTT cell viability test, the concentration of ZnO nanoparticles up to 0.1 mg/mL was biocompatible and had no major and significant damaging effect to the human cells. Ag/ZnO nanoparticles exhibited the best antimicrobial activity and highest shear bond strength.	37/39
Kim et al. 2018 [36]	National Research Foundation of Korea (NRF) grant funded by the Korea government (MSIP) (NRF2015R1C1A1A01051832).	Human gingival fibroblasts (HGFs; ATCC, Manassas); *S. mutans* (Ingbritt); N = 60 human premolars for 6 groups for bracket SBS and ARI index	Orthodontic bonding agents containing zinc-doped bioactive glass (BAG; orthodontic bonding agents containing BAG were prepared by mixing BAG with flowable resin; resin disk specimen	TransbondTM XT (TXT, 3M) and CharmfilTM Flow (CF, A2 shade, Denkist)	Mechanical and biological properties: Ion release, cytotoxicity, antibacterial properties, microhardness, shear bond strength, adhesive remnant index (ARI), and micro-computed tomography was performed after pH cycling to analyze remineralization.	Bonding agents with zinc-doped BAG have stronger antibacterial and remineralization effects compared with conventional adhesives (*p* < 0.05).	38/39
Zeidan et al. 2022 [57]	The Science, Technology, and Innovation Funding Authority (STDF) in cooperation with The Egyptian Knowledge Bank (EKB)	*S. mutans* strain (ATCC 25175)	Brackets coated with nanoparticles of Ag, ZnO, and a combination of both Ag/ZnO; N = 48 brackets, stainless steel “American orthodontics 0.018′’ slot size of lower premolars	Brackets as received without modifications	Antibacterial activity on *S. mutans* and *L. acidophilus* using CFU, evaluated immediately after coating and after 3 months.	Combination of silver and zinc oxide nanoparticles had the highest bacterial reduction. The coating of orthodontic brackets could be further assessed in clinical application to prevent decalcification.	37/39
**Other Aspects**							
Chen et al. 2021b [55]	N/A	Human gingival fibroblasts (hGF) cells	Pure Zn used in barrier membrane in GBR therapy	None	Degradation behavior in artificial saliva solution, cytotoxicity, and antibacterial activity investigated to explore Zn degradation and associated biocompatibility in the case of premature membrane exposure.	Zn degradation rate in artificial saliva was 31.42 μm (year-1) after 28 days of immersion. Zn presented an acceptable HGF cytocompatibility and a high antibacterial activity against *P. gingivalis*, exhibiting appropriate degradation behavior, adequate cell compatibility, and favorable antibacterial properties in the oral environment.	36/36
Mishra et al. 2020 [47]	N/A	Human dental pulp stem cells and human red blood corpuscles, *Staphylococcus aureus* (ATCC 9144)	Biomaterial composed of zinc–carboxymethyl chitosan(CMC)–genipin synthesized and transformed to porous scaffolds using freeze drying method; the scaffolds were cross-linked and stabilized with genipin and zinc (2 M zinc acetate)	Redundant controls	FTIR spectroscopic data, scanning electron microscopy, compressive strength, biodegradation, and antibacterial properties.	The scaffolds seemed to support the adhesion and proliferation of human dental pulp stem cells and were hemocompatible with human red blood corpuscles. The scaffolds were found to be antibacterial and mildly antibiofilm against *S. aureus*.	32/36
Wiesmann et al. 2021 [56]	Max Planck Graduate Center, Mainz, Germany, and by the research focus group “BiomaTiCS—Biomaterials, Tissues and Cells in Surgery” of the University Medical Center, Mainz, Germany	Human gingival fibroblasts (Provitro AG, HFIB-G) and human umbilical vein endothelial cells (HUVECs) isolated from human umbilical cord veins	Incorporation of zinc oxide nanoparticles 20 nm (ZnO NPs) in biomaterials for tissue regeneration; zinc oxide nanoparticles were obtained from IoLiTec Ionic Liquids Technologies GmbH (Product Nr.: NO-0011-HP)	None	Antibacterial effects and metabolic activity on fibroblasts and endothelial cells, and biocompatibility with chicken chorioallantoic membrane assay (CAM)l flow cytometry for cell death; cell viability assay for cellular metabolic activity (CMA) with the alamarBlue™ Cell Viability Reagent.	ZnO NPs had favorable properties for biomaterials modification and could help to guide the tissue reaction and promote complication-free healing (*p* ≤ 0.001).	35/39

#### 3.2.1. Conservative and Cariology

Two studies applied Zn in dental adhesive containing zinc dimethacrylate ionomer (ZDMA) [38] and ZnO [50], one study used an Ag/ZnO nanocomposite [42], and one study used elastomeric temporary resin-based filling materials with zinc methacrylate [32]. Barma et al. used a ZnO-NP varnish [49], Lavaee et al. used zinc sulfate and zinc acetate [30], and Suzuki et al. used metal chlorides (ZnCl_2_) and metal acetates ((CH_3_COO)_2_Zn) [33]; while one study used solutions prepared with nano ZnO powder [39], one study used solutions prepared with zinc-doped nanoparticles (Zn-NPs) [13], one study used solutions prepared with divalent cation Zn^2+^ alone and in combination with ZnCl [43], and one study used solutions prepared with zinc-substituted hydroxyapatite/alendronate-grafted polyacrylic acid hybrid nanoneedles [44]. Manus et al. applied a dentifrice with zinc citrate formulations with increasing replacements of zinc oxide (a water-insoluble source of zinc ions) to generate a dual-zinc active system [31], and Tabatabaei et al. applied zinc toothpastes and mouthwashes [40].

##### Antibacterial Properties

Five articles evaluated the antibacterial effects on *S. mutans*: growth [38,49], biofilm, acid production, and antioxidant potential [49]; biofilm formation, virulence factors, and related genes expressions at sub-minimum inhibitory concentrations (MIC) [42]; inhibitory and bactericidal effects, MIC, and minimum bactericidal concentration (MBC) [30]; and biofilm formations and growth and EPS extraction [43]. Eskandarizadeh et al. reported an excellent antimicrobial property against *S. mutans* [38]. According to Barma et al. [49], a significantly suppressed expression of *S. mutans*’ virulence-factors-related genes was observed [42]. Zinc sulfate and zinc acetate salts had an inhibitory effect on *S. mutans*, and zinc sulfate solution caused higher MIC and MBC compared to penicillin and chlorhexidine [30], while Steiger et al. reported strong effects but no significance [43].

One article studied bactericidal properties via a biofilm model after the formation of an acquired pellicle, considering metabolic activity and live/dead staining, highlighting how ZnO adhesive caused lower CFU/mL not only in *S. mutans* but also in other microorganisms with lower metabolic activity and higher dead bacteria [50]. One article evaluated growth, MIC, and MBC on *S. mutans*, *E. faecalis*, *L. fermentum*, and *C. albicans*, observing the greatest antimicrobial effect against *S. mutans* and *E. faecalis* [39]. In another article, the antimicrobial effects on *E. faecalis* and a *S. mutans* were studied [32]. The inhibitory effects on bacterial growth were also analyzed according to Zn ions’ binding to gaseous hydrogen sulfide (H_2_S) and their minimum concentration required to inhibit H_2_S volatilization, and Suzuki et al. showed how zinc ions’ effect on the growth of oral bacteria was strain-dependent (*F. nucleatum* was the most sensitive) [33]. There was also an inhibitory effect on oral malodor via direct binding with gaseous H_2_S and a volatile sulfur compound-producing oral bacteria suppression [33]. Zn bioavailability enhancement, penetration and retention, and antibacterial efficacy were also considered, and Manus et al. highlighted a significant bacterial metabolic activity reduction related to bacterial glycolytic function and total oxygen consumption [31].

##### Cytotoxicity

Cytotoxicity was analyzed in four studies [32,40,44,49]. Tabatabaei et al. reported a time- and concentration-dependent cytotoxic effect of <50% for all zinc lactate concentrations [40], Barma et al. confirmed ZnO-NP low cytotoxicity [49], and Xu et al. showed a cell viability higher than 80% [44]. Other biological properties investigated were biocompatibility [32], cariogenic dissolution of hydroxyapatite [43], and dentin hypersensitivity, considering how Zn-nanoparticles treatment caused higher complex modulus values to be attained in intertubular and peritubular dentin [13].

##### Physical, Chemical, and Mechanical Properties

The physical, chemical, and mechanical properties analyzed were degree of conversion (DC) in three studies [32,38,50]; Zn ion release amount, compressive strength, and shear bond strength in only one study, which observed that compressive strength was significantly reduced and dentine shear bond strength significantly higher for 5% ZDMA [38]; flexural strength and elastic modulus in only one paper, in which ZnO incorporated at 7.5% caused lower flexural strength, while at 2.5%, it had a greater modulus of elasticity [50]; and microleakage, water sorption/solubility, depth of cure, ultimate tensile strength, and Shore D scale hardness in one study, which reported lower microleakage and water sorption and higher ultimate tensile strength values [32].

#### 3.2.2. Endodontics

Four articles analyzed zinc properties related to the endodontic aspect [12,28,34,45]. These articles evaluated the antibacterial effects of zinc considering its application on endodontics medications [34], endodontic obturation material [28], or experimental adhesive resin [12,45]. Other biological properties studied were cytotoxicity [12,28], biocompatibility, radiopacity, and pH [28]. The mechanical properties analyzed were degree of conversion, ultimate tensile strength, and hardness and softening in solvent [45].

Zinc showed strong antibacterial properties in all the formulations used: in combination with Ag [34] or incorporated into endodontic cements [28] or adhesives such as zinc oxide quantum dots (ZnOQDs) [12] or zinc chloride (ZnCl_2_) [45]. In particular, the combination with silver revealed a powerful effect against planktonic and biofilm-resident *E. faecalis* related to the depolarization of the bacterial membrane with the addition of Zn^2+^. Garcia et al. also reported an antibacterial activity against biofilm formation and planktonic bacteria, which did not affect the adhesives’ physicochemical and mechanical properties [45]. Furthermore, no differences, in terms of biological and physical properties, were reported when comparing zinc oxide eugenol cements to other endodontic cements (MTA) [28].

#### 3.2.3. Periodontology and Implantology

Periodontal and implantological fields were considered in five studies [35,46,51,52,53], which analyzed zinc as a coating metal with a strontium nanorod for sandblasted and acid-etched titanium dental implants [51], a biocomposite coating comprising ZnO and chitosan deposited on a porous Ti oxide [52], zinc oxide nanoparticles on implant connection [53], glucose-1-phosphate biofunctionalized zinc peroxide nanoparticles [46], and polymeric PolymP-n Active nanoparticles [35]. Antibacterial activity was evaluated in four studies [35,46,52,53], but the following were also investigated: osteogenesis and inhibition of biofilm formation [51]; wettability, anti-scratch ability, bioactivity, and corrosion resistance [52]; cytotoxic effects [53]; and nano-roughness surface [35].

Zinc improved antibacterial properties when used as a coating metal [52], promoted an inhibitory effect on Gram-negative anaerobes and on *A. actinomycetemcomitans* [46], and suppressed *S. aureus* biofilm formation [51], but reductions in other bacterial viabilities were more pronounced with other metal nanoparticles [35]. While Vergara-Llanos et al. [53] highlighted how zinc antibacterial activity in a monospecies model was strain-dependent and induced mitochondrial dose-dependent cytotoxicity, increasing LDH release and intracellular ROS generation [53], zinc also promoted initial cellular adhesion, proliferation, and osteogenic differentiation [51], the critical force of scratching and corrosion resistance in association with cytocompatibility [52], and good roughness properties [35].

#### 3.2.4. Orthodontics

Three articles analyzed the properties of zinc related to orthodontics aspects [36,54,57]. These articles evaluated zinc properties considering its application on composite resin with silver/zinc oxide nanoparticles [54], bonding agents containing zinc-doped bioactive glass [36], and brackets coated with nanoparticles of ZnO or Ag/ZnO [57]. The biological properties studied were ion release, cytotoxicity, biocompatibility, and antimicrobial properties [36,54], as well as antibacterial activity on *S. mutans* and *L. acidophilus* [57].

Zinc has a significant antimicrobial effect and biocompatibility [54] and a strong antibacterial effect [36]. Ag/ZnO in combination caused a great reduction on *S. mutans* and *L. acidophilus* [57]. Kim et al. also reported a remineralization effect [36]. Wettability [54], microhardness, and adhesive remnant index [36], as well as shear bond strength [36,54], were also analyzed, with Ag/ZnO nanoparticles exhibiting high shear bond strength values [54].

#### 3.2.5. Other Aspects

Three articles studied zinc effects when applied in biomaterials [47,55,56], such as a barrier membrane in GBR therapy [55], a scaffold with zinc–carboxymethyl chitosan [47], and a biomaterial for tissue regeneration with zinc oxide nanoparticles [56]. The authors of these studies considered the effect of zinc composition related to antibacterial activity [47,55,56], biodegradation [47,55], biocompatibility [55,56], behavior in artificial saliva and cytotoxicity [55], compressive strength [47], and metabolic activity on fibroblasts and endothelial cells [56].

Zn exhibited adequate degradation behavior and cell compatibility and favorable antibacterial properties in the oral environment [55]. Zinc scaffolds were compatible to human red blood corpuscles and had mildly antibacterial and antibiofilm effects against *S. aureus* [47]. Furthermore, ZnO nanoparticles showed good properties for biomaterials modification and could promote tissue reaction and complication-free healing [56].

Table 5 summarizes the characteristics of the evaluated zinc dental products and their main effects.

## 4. Discussion

The results highlighted in the literature and the data obtained in this review show that zinc is indispensable for oral health and therefore an element found in many dental materials, not only those destined for home hygiene. In fact, zinc is increasingly being used to make modern, affordable, and valuable materials for the prevention of oral diseases. Zinc is effective in the prevention and treatment of aphthous ulcers, dental caries [58,59], and periodontal disease [60]. It promotes remineralization and inhibits the dissolution of dental hard tissues. It has also been identified as a biomarker of oral squamous cell carcinoma and should therefore be evaluated for suspicious lesions [58]. It would therefore be crucial to increase knowledge of the role of this element and the outcomes of its deficiency, considering the naturally available food sources for a healthy and complete diet and how other elements compete for its absorption, such as iron supplements. In vulnerable individuals, such as the immunocompromised, patients undergoing chemo- or radio-therapy, patients with oral disease, pregnant women, and children, it is therefore extremely important. Certainly, zinc deficiency could lead to the onset of oral pathologies; however, since there are very few evaluable studies in the literature [1], before a clinical protocol of zinc’s supplementation as a co-adjuvant therapy to specific treatments can be established, further in vivo research with adequate patient sample sizes and long-term follow-up is needed.

Thus, this review aims to provide an overview of the properties and characteristics of zinc in various dental fields and its biological effects in the oral cavity, examining studies on healthy and unhealthy patients or on pathological or non-pathological cell populations.

The null hypothesis was rejected; the addition of zinc to dental materials did not affect biological and mechanical properties.

In this review, we included only five in vivo studies. Interestingly, all authors focused on the antibacterial power of zinc, albeit using different materials and therefore being difficult to compare. It is obviously evident how important it is to minimize the number of pathogenic bacteria present inside the oral cavity during daily hygiene practices using the mechanical action of brushing combined with the chemical action of zinc, but how clinically important bacterial control is during endodontic therapies it is also evident given the impossibility of achieving complete sterility of the endo canal system, especially in association with periapical lesions. It would be interesting to investigate this topic with further studies by increasing the numerosity and homogeneity of the sample under consideration, and also taking into account the adult population since, with regard to the use of zinc in endodontic medications, only pediatric subjects have been considered, and there are no similar studies analyzing the same properties in adult patients at present.

In vitro evaluation of materials used in conservative therapy and endodontics containing zinc also showed that the main property examined by the authors was zinc’s antibacterial capacity. Zinc demonstrated excellent effects in all the formulations examined, although their therapeutic applications and clinical modalities of use are so different from each other that it is a complex matter to perform an accurate comparison. The wide variability in the studies is also reflected in what is clinically expected from the reconstructive or three-dimensional endodontic medication and obturation materials, respectively, although the possibility of having an effective antibacterial effect is certainly the common element in both. The variability in the different types of tests used to analyze the properties under investigation is also a direct consequence of the heterogeneity of the studies. The addition of zinc does not seem to affect the activity of the materials, and there is no evidence of negative mechanical performance [45]; indeed, in some cases, an improvement has been shown, with lower microleakage and water sorption and higher ultimate tensile strength [32]; however, this is an aspect to be given more consideration in future studies as high mechanical performance is required for such materials, especially when used in pedodontics, and it would be crucial to combine a good antibacterial effect with improved mechanical features.

Nanotechnology used in conservative dentistry is an integral part of the latest dental research. ZnO NPs are biocompatible and possess antimicrobial action against a wide range of microorganisms. The antibacterial properties appear to increase with increasing particle surface area, but issues regarding toxicity and particle release patterns, as well as long-term properties, still need to be resolved. Unfortunately, particle dispersion could increase flexural strength, decrease shear strength, and decrease compressive strength in composite materials [61,62].

In contrast, the application of zinc in periodontology has led to results with obvious discordance, even considering the antibacterial aspect. Materials with heterogeneous chemical compositions and different clinical uses have certainly been used, but all authors have evaluated the use of zinc as a coating material or for implant/prosthetic use. In fact, in some studies, zinc has been observed to have improved antibacterial properties [46,51,52], while Sánchez et al. have found that the reduction in bacterial viability was more pronounced with the use of other types of metal nanoparticles [35]. Other recent reviews also agree on the antibacterial properties of zinc [63,64]. Zn used in various products has been shown to be effective against common prevalent oral health problems, such as dental caries, gingivitis, and periodontitis, as it exhibits good oral substantivity and is retained in plaque and saliva for many hours; also, with repeated applications, there is a build-up effect in plaque [3]. Almoudi et al. [63] reported significant efficacy in terms of the inhibition of *S. mutans* growth by zinc, even at low concentrations, indicating that it can be used safely in oral products. However, Zn has low enamel remineralization potential, and no net effect on caries has been reported with in vitro studies [3]. Griauzdyte et al. [64] supported the antibacterial action of zinc against periodontal pathogenic bacteria, confirmed by Liu et al. [65], suggesting that zinc may play a key role in immune defense, inflammatory responses, and bone remodeling, and that its homeostasis is essential for periodontal regeneration. The integration of zinc within an appropriate dosage range or the regulation of zinc transport proteins could potentially enhance regeneration by boosting immune defenses or promoting cell proliferation and differentiation [65].

Regarding cytotoxicity, Vergara-Llanos et al. [53] pointed out how zinc induces dose-dependent mitochondrial cell toxicity, while Lin et al. noted how zinc is cytocompatible [52]. Furthermore, denture adhesives with zinc could be responsible for decreased cell viability, ROS production, aberrant cell morphologies, and the induction of apoptosis and cell death [66]. Zn cytotoxicity was also found in ZOE-based dental materials during the initial setting phase with immortalized human dental pulp stem cells [67] and three-dimensional (3D) cultures of immortalized human oral keratinocytes [68]; such moderate or severe cytotoxic reactions appear to be related to the constant dissolution of such materials when exposed to an aqueous environment for prolonged periods. Another study suggested that in a model of non-keratinized oral epithelial cells at concentrations exceeding 0.031% *w*/*v*, ZnO was the most cytotoxic nanomaterial among several common oral hygiene products because nano-size effects have some impact on the cytotoxicity of a material [69]. Significant toxic effects of ZnO-NP were also found at concentrations of ~50–100 μg/mL on human periodontal ligament fibroblasts, depending on the concentration and duration of exposure [70]. Although it should be noted that the authors used different types of assays, materials, and cells, it would be interesting to understand whether it is zinc alone that causes cytotoxicity or whether it is because it is incorporated into other dental materials. The same problems also occurred from the orthodontic point of view since the chemical formulations of the materials studied and their use were very heterogeneous, as were the biological and mechanical properties and the tests used for their evaluation; as a result, it is extremely difficult to make realistic comparisons between the various studies.

### Limitations of the Study

QuADs allows for the qualitative analysis of selected papers with reliability, but there are some limitations to be taken into consideration. We extended the investigation of zinc’s effects to both in vivo and in vitro studies, including both healthy human or pathological cell populations, to obtain a complete overview of zinc’s biological and mechanical effects. Comparison of these studies, although reliable, is difficult; in the case of the in vivo studies, this difficulty can be due to the enrollment of a minimal sample size or non-homogeneous patient populations, especially with regard to age. In addition, the chemical compositions of the materials used are extremely variable, both in terms of the types of materials and the percentages of zinc used, either isolated or bonded to other components, which certainly influence the overall effect. It will also be important to carry out further research taking into consideration some of the important limitations found in this study, such as the inclusion of work in languages other than English and Italian and the use of new materials in innovative fields of dentistry dedicated to patients with special needs or to fragile or geriatric patients. In the literature, there are still few in vivo studies, and there are no studies on 3D tissue or organoid models. It is important to obtain more information regarding zinc in in vivo conditions in order to precisely define its therapeutic use in various formulations.

## 5. Conclusions

It is clear that the property of greatest interest in the use of zinc is its capacity to achieve an antibacterial effect; however, further investigation is needed in order to establish precise adjuvant zinc therapies in patients with oral disease.

In consideration of the results obtained and the limited in vivo material in the literature on this subject, from this review comes the intention of the research group to set up a collateral in vivo study to analyze the efficacy of plaque removal by establish a comparison between toothpaste containing zinc and fluoride and standard fluoride toothpaste.

## Figures and Tables

**Figure 1 materials-17-00800-f001:**
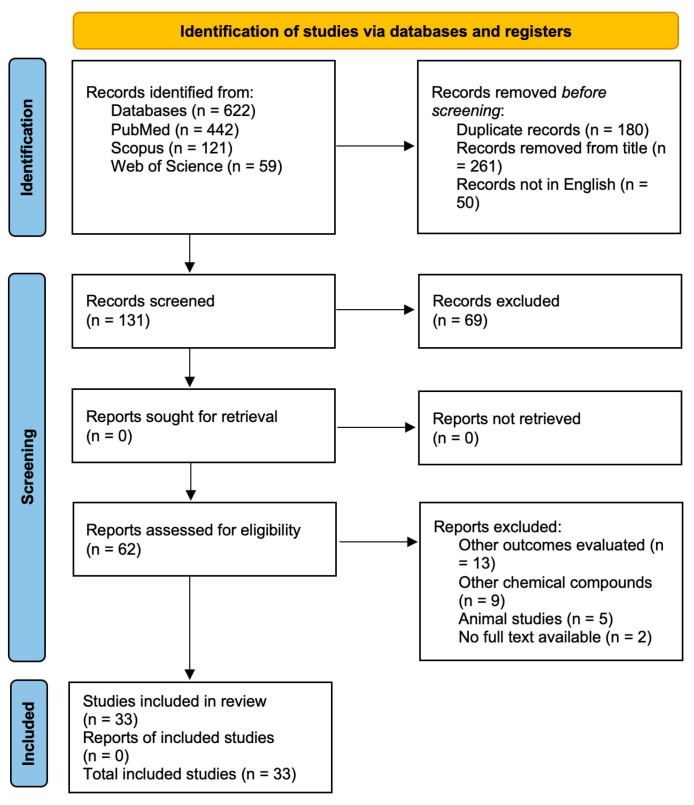
Flow diagram of the screening and selection process according to the PRISMA statement.

**Table 1 materials-17-00800-t001:** Research question with the PICOS strategy format and inclusion/exclusion criteria.

PICOS	
Population/patient	Children and adult patients
Intervention/indicator	Zinc exposure
Comparator/control	No zinc exposure
Outcomes	Oral health
Study design	In vivo (clinical trials: controlled clinical trials (CCTs) and randomized controlled trials (RCTs); and observational studies: case control and cohoort studies) and in vitro
Other inclusion criteria	English or Italian studies published in the last 10 years
Exclusion criteria	Studies on other chemical compounds or with no full-text available, systematic reviews, animal studies, case reports, editorials, opinions, surveys, guidelines, conferences, commentaries

**Table 2 materials-17-00800-t002:** Strategy search.

**Pubmed** (2 March 2022)
Query
((“zinc”[MeSH Terms] OR “zinc”[All Fields]) AND (“anti bacterial agents”[Pharmacological Action] OR “anti bacterial agents”[MeSH Terms] OR (“anti bacterial”[All Fields] AND “agents”[All Fields]) OR “anti bacterial agents”[All Fields] OR “antibacterial”[All Fields] OR “antibacterials”[All Fields] OR “antibacterially”[All Fields]) AND (“mouth”[MeSH Terms] OR “mouth”[All Fields] OR “oral”[All Fields])) NOT (Systematic Review [Publication Type] OR Review [Publication Type] OR Meta-Analysis [Publication Type] OR Comment [Publication Type] OR Congress [Publication Type] OR Editorial [Publication Type] OR Case Reports [Publication Type] OR Clinical Conference [Publication Type] OR Comment [Publication Type] OR Consensus Development Conference [Publication Type]) Filters: English, Italian, 10 years.
**Scopus** (2 March 2022)
Query
(TITLE-ABS-KEY (“zinc”) AND TITLE-ABS-KEY (“anti bacterial agent*” OR “antibacterial*”) AND TITLE-ABS-KEY (“mouth” OR “oral”) AND NOT TITLE-ABS-KEY (systematic AND review AND [doctype] OR review AND [doctype] OR meta-analysis AND [doctype] OR comment AND [doctype] OR congress AND [doctype] OR editorial AND [doctype] OR case AND reports AND [doctype] OR clinical AND conference AND [doctype] OR comment AND [doctype] OR consensus AND development AND conference AND [doctype])) AND PUBYEAR > 2011 AND PUBYEAR < 2023 AND (LIMIT-TO (DOCTYPE, “ar”))
**Web of Science** (2 March 2022)
Query
TS = (((“zinc”) AND (“anti-bacterial agents” OR “anti-bacterial*” OR “antibacterial*”)) AND (“mouth” OR “oral”)) AND LANGUAGE: (English OR Italian) AND DOCUMENT TYPES: (Article) Timespan = 2012–2022

**Table 5 materials-17-00800-t005:** Types of products used with zinc and their properties.

Dental Products	Zinc Advantages
**In vivo studies**	
Zinc oxide-ozonated oil (ZnO-OO) (0.007 cc Ozonil, Ozone Forum of India)	Pulpectomy agent
Zinc-substituted carbonated hydroxyapatite dentifrice (BioRepair, Wolff)	Antimicrobial activity in co-adiuvant periodontal therapy
Zinc oxide eugenol paste (ZOE) (Biodynamics)	Endodontic lesion sterilization and tissue repair
New fluoride toothpastes with Dual Zinc plus Arginine formulations (Dual Zinc plus Arginine; Colgate-Palmolive Company)	Antibacterial activity
Herbal toothpaste incorporating zinc	Antibacterial activity
**In vitro studies**	
**Conservative and Cariology**	
Zinc oxide nanoparticles synthetized (ZnO-NP) varnish	Antimicrobial, bactericidal, and antioxidant activity
Dental resin adhesive with zinc dimethacrylate ionomer (ZDMA)	Antibacterial activity, higher dentine shear bond strength, and lower compressive strength
ZnO incorporated in dental adhesive	Antibacterial activity, higher degree of conversion, and elastic modulus
Ag/ZnO nanocomposite	Antimicrobial activity and suppression of virulence factors related genes expression
Zinc sulfate (Merck) and zinc acetate (Falcon) solutions	Antibacterial activity
Dual Zinc active system dentifrice, with zinc citrate and zinc oxide	Antibacterial activity
Solutions with nano ZnO (nZnO) powder (Research Nanomaterials Inc.)	Antimicrobial activity
Elastomeric temporary resin-based filling materials with zinc methacrylate (ZM, Aldrich Chemical Co.)	Antibacterial activity, lower microleakage and water sorption, and higher ultimate tensile strength
Divalent cation Zn^2+^ and in combination ZnCl, ZnCl_2_	Antibacterial activity
metal chlorides, ZnCl_2_, and metal acetates, (CH_3_COO)_2_Zn	Antimicrobial activity
toothpastes and mouthwashes with sodium fluoride, sodium lauryl sulfate, cocamidopropyl betaine, zinc lactate, paraben, and sodium benzoate	Cytocompatibility
Zinc-doped NPs (Zn-NPs), PolymP-n Active nanoparticles (NanoMyP): aqueous solutions of ZnCl_2_	Higher intertubular and peritubular complex modulus values
Zinc-substituted hydroxyapatite/alendronate-grafted polyacrylic acid hybrid nanoneedles (ZHA@ALN-PAA) of Zn(CH_3_COO)_2_%2H_2_O	No cytotoxcity
**Endodontics**	
Ag^+^-Zn^2+^ atomic combination ratios	Antibacterial activity
Zinc oxide quantum dots (ZnOQDs) into adhesive resin	Antibacterial activity
Zinc-based particle with ionic liquid as filler for experimental adhesive resin	Antibacterial activity and no pulp cytotoxcity
Zinc oxide eugenol (ZOE), Vitapex, and Calen paste thickened with zinc oxide (ZO)	Biocompatibility and antimicrobial activity
**Periodontology and Implantology**	
co-incorporated zinc and strontium nanorod coating on sandblasted and acid-etched titanium (SLA-Zn/Sr)	Good interface bonding strength; enhanced corrosion resistance, biofilm inhibition, and cellular initial adhesion; and proliferation and osteogenic differentiation promotion
Glucose-1-phosphate (Glc-1P) biofunctionalized zinc peroxide (ZnO_2_) nanoparticles	Antimicrobial activity
Implant with inner layer of nanoporous TiO_2_ and outer layer of chitosan matrix with ZnO nanoparticles	Antibacterial activity, bioactivity, cytocompatibility, and improved corrosion resistance
Hydroxyapatite discs coated with nanoparticles (NPs) doped with zinc	Antibacterial activity and icrements in roughness
ZnO-NPs	Antibacterial activity and biocompatibility
**Orthodontics**	
New bioactive orthodontic composite resin containing silver/zinc oxide (Ag/ZnO) nanoparticles	Antimicrobial activity, biocompatibility, and higher shear bond strength
Orthodontic bonding agents containing zinc-doped bioactive glass (BAG)	Antimicrobial activity and remineralization effects
Brackets coated with nanoparticles of Ag, ZnO and a combination of both Ag/ZnO	Antibacterial activity
**Other Aspects**	
Pure Zn used in barrier membrane in GBR therapy	Antibacterial activity and cytocompatibility
Biomaterial composed of zinc-carboxymethyl chitosan (CMC)-genipin synthesized and transformed to porous scaffolds	Antibacterial activity, good adhesion and proliferation of human dental pulp stem cells, and hemocompatibility
Incorporation of zinc oxide nanoparticles (ZnO NPs) (IoLiTec Ionic Liquids Technologies GmbH) in biomaterials for tissue regeneration	Antibacterial activity, tissue reaction, and complication-free healing promotion

## Data Availability

The original contributions presented in the study are included in the article/Appendix A, further inquiries can be directed to the corresponding author/s.

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
