# Peer review of "Systematic Review of Zinc’s Benefits and Biological Effects on Oral Health"

_materials, 2024, doi:10.3390/ma17040800_

Round 1

Reviewer 1 Report (Previous Reviewer 4)

Comments and Suggestions for Authors

Dear author, I have no more questions. Thank you. 

Author Response

We thank the reviewer for the positive opinions.

Reviewer 2 Report (New Reviewer)

Comments and Suggestions for Authors

The topic is interesting, the study, review is well done. The used methodology prove the profound character of the research, the used methodology is well-thought-out. One of the forte part is the critical comments of the authors and the comparative analyses of the papers.

There are 2 small remarks:

line 56: hyphenation is wrong

line 239: the authors should give more concrete details about the mentioned mechanical properties

Overall, at the and the authors should make a summary table regarding the obvious advantages and/or disadvantages of zinc.

Author Response

We thank the reviewer for the positive opinion.

line 56: hyphenation is wrong

We are really sorry but we are not able to correct the hyphenation, because it is automatic with justified text format.

line 239: the authors should give more concrete details about the mentioned mechanical properties

Thanking the reviewer for the note, we have included more information as requested, as follows: " flexural strength and elastic modulus in only 1 paper, in which ZnO incorporated at 7.5 % caused lower flexural strength, while at 2.5 % had a greater modulus of elasticity [50];"

Overall, at the and the authors should make a summary table regarding the obvious advantages and/or disadvantages of zinc.

We apologize for the misunderstanding and thank the reviewer for the comment; Table 5, with a summary of zinc properties and the related types of zinc products used, already included the advantages of the substance, has been modified to enhance the requirements as requested by the reviewer. In the studies considered, the disadvantages were not significant to be reported.

Reviewer 3 Report (New Reviewer)

Comments and Suggestions for Authors

Dear authors, thank you for submitting the manuscript "Systematic review of zinc benefits and biological effects on oral health".

I read you article and here is my feedback:

Introduction. It will be good to mention that some dental implants have incorporated zinc into the titanium implant surface.

Make a small table with summary for inclusion and exclusion criteria.

Materials and Methods. For other aspects, please also mention if there is material/techniques employed specially in geriatric dentistry for elder patients.

Also you missed products used in pediatric dentistry, please mention them (missing in table as well).

I see several limitations in this study, please include more of them such as using more languages in the search, new materials used and new branches of dental care such as special patients, geriatric patients, etc.

Author Response

Introduction. It will be good to mention that some dental implants have incorporated zinc into the titanium implant surface.

We thank the reviewer for the note and added a specifications in the introduction section where we mentioned the implants, as follows: “…dental implant with surface coated in zinc…”

Make a small table with summary for inclusion and exclusion criteria.

We thank the reviewer for the idea and added a small summary for inclusion and exclusion criteria adding the elements that were missing from materials and methods section in table 1.

Materials and Methods. For other aspects, please also mention if there is material/techniques employed specially in geriatric dentistry for elder patients.

We thank the reviewer for pointing out this issue, in the papers we evaluated and included, the adult population considered had no age limits, unfortunately, however, no papers were found that specifically looked only at geriatric patients; We did however include among the in vivo studies the papers by Hagenfeld et al. 2019, and Sreenivasan et al. 2020 which also considered patients up to 75 and 63 years of age respectively, investigating the application of toothpaste with zinc.

Also you missed products used in pediatric dentistry, please mention them (missing in table as well).

We apologize for the misunderstanding, we have also included paediatric patient populations in our research, in fact in vivo studies by Doneria et al. 2017 and Moura et al. 2021, evaluated endodontic tissue repair and lesion sterilization after treatments with zinc oxide-ozonated oil in children with 4-8 years, and zinc oxide eugenol paste (ZOE) in children with mean age of 5.5 years, respectively.

I see several limitations in this study, please include more of them such as using more languages in the search, new materials used and new branches of dental care such as special patients, geriatric patients, etc.

Thank you very much to the reviewer for the proposal, we understand the reasons, and we have proceeded to add a specific paragraph in the "study limitations" section as requested: "It would also be important to carry out further research taking into consideration some important limitations found in this study, such as the inclusion of work in languages other than English and Italian, the use of new materials in innovative fields of dentistry dedicated to patients with special needs, fragile or geriatric patients.”

Reviewer 4 Report (New Reviewer)

Comments and Suggestions for Authors

Zinc is a very important microelement needed to maintain health in the oral cavity. Therefore, every publication in this area is very important. This topic is very extensive and difficult to cover in one article, so my appreciation goes to the authors.

However, as a reviewer, I have a few comments:

Abstract

What does the RCTS abbreviation mean? If you use abbreviations, explain them the first time you use them, so that the reader does not have to look for them and has no doubts about what he is reading. Thank you

Introduction

 Line 60                                                                .

anti-MMP activity- please explain this aberration.

It would be good to add a thesis in the introduction part. For example, the use of zinc addition to dental materials does not affect their biological and mechanical properties. Or something like that.

 Materials and methods

Figure 1

It will be better to use traditional  PRIZMA diagram with block, better visibility.

Results

Line 135

Honestly, removing 261 typical articles after reading their title without an abstract is a bit of an imprecise criterion for selecting the work, but it's your choice. I would also prefer to read abstract, because sometimes the title itself is not very informative.

Line 147

Of the in vitro studies, 16 used human cell populations, considering the gastro-intestinal apparatus using liver cancer cell line [49], in particular for the orofacial district were 1analyzed gingival fibroblasts [36, 40, 53-56], oral keratinocyte cells [44], pulp fibroblasts 1[12] or dental pulp cells [28, 45,47], and epithelial tissue and saliva-derived biofilm models-

This sentence it will be good to improve for example.  

After omitting in vitro work, the following studies on the effects of zinc were distinguished from among 16 studies on humans: gastro-intestinal apparatus using liver cancer cell line [49], in  orofacial district were analyzed gingival fibroblasts [36, 40, 53-56], oral keratinocyte cells [44], pulp fibroblasts 1[12] or dental pulp cells [28, 45,47], and epithelial tissue and saliva-derived biofilm models.

Or instead of listing it all, I would use a 2-column table: place of activity, publications, but also including the author number of the publication. It would be more readable.

Table 3

It will be good to add

Doneria  et al; 2017

Line 216

MIC and MBC- please explain these aberrations

sulfide (H2S- sulfide (H2S

I would divide the3.2.1. Conservative and cariology  paragraph into even smaller paragraphs: e.g. odor, cytotoxicity, physical properties of materials after the addition of Zn, etc., it would be more readable.

  I very often refer to review works and read only those paragraphs that I am most interested in, to make it easier to find source publications. This precise division of information into subheadings makes reading very easy.

Line 254

Ag revealed- in previous sentence you used Ag, now you can write silver .

E. faecalis- E. faecalis

Zn2+,- Zn2+. Garcia et al

Discussion

Line 361

especially when used in pedodontics, and it would be crucial to combine a good antibacterial-  or periodontics?- The entire article concerns dentistry, so the conclusions will not include the effect of zinc on feet?

Line 374

have evaluated the use of zinc as a coating material for periodontal or implant-prosthetic

use.  Costing material for  implant I agree, but for the surface of the root this is the new info for me, please add ref to it.

Line 383

S. mutans- S. mutans

Discussion well written, I like it!

Comments on the Quality of English Language

small improvements needed

Author Response

Abstract

What does the RCTS abbreviation mean? If you use abbreviations, explain them the first time you use them, so that the reader does not have to look for them and has no doubts about what he is reading. Thank you

 We are sorry for the misunderstandings, the abbreviations are for: controlled clinical trials (CCTs), and randomized controlled trials (RCTs);

We have corrected and explain them the first time they appear both in the abstract and in the text.

Introduction

 Line 60                                                                .

anti-MMP activity- please explain this aberration.

We thank the reviewer to have pointed out the mistake and change anti-MMP activity to anti-matrix metalloproteinase activity at line 60

It would be good to add a thesis in the introduction part. For example, the use of zinc addition to dental materials does not affect their biological and mechanical properties. Or something like that.

We thank the reviewer for the proposal and add the null hypothesis at the end of the introduction as follows: “Null hypothesis: the addition of zinc to dental materials affected biological and mechanical properties.”, and confuted in the discussion section: “The null hypothesis was rejected, the addition of zinc to dental materials did not affect biological and mechanical properties.”

 Materials and methods

Figure 1

It will be better to use traditional  PRIZMA diagram with block, better visibility.

We modified figure 1 as recommended.

Results

Line 135

Honestly, removing 261 typical articles after reading their title without an abstract is a bit of an imprecise criterion for selecting the work, but it's your choice. I would also prefer to read abstract, because sometimes the title itself is not very informative.

We thank the reviewer for this note, we understand the motivation of the comment and will take it very much into account for future work.

Line 147

Of the in vitro studies, 16 used human cell populations, considering the gastro-intestinal apparatus using liver cancer cell line [49], in particular for the orofacial district were 1analyzed gingival fibroblasts [36, 40, 53-56], oral keratinocyte cells [44], pulp fibroblasts 1[12] or dental pulp cells [28, 45,47], and epithelial tissue and saliva-derived biofilm models-

This sentence it will be good to improve for example.  

After omitting in vitro work, the following studies on the effects of zinc were distinguished from among 16 studies on humans: gastro-intestinal apparatus using liver cancer cell line [49], in  orofacial district were analyzed gingival fibroblasts [36, 40, 53-56], oral keratinocyte cells [44], pulp fibroblasts 1[12] or dental pulp cells [28, 45,47], and epithelial tissue and saliva-derived biofilm models.

Or instead of listing it all, I would use a 2-column table: place of activity, publications, but also including the author number of the publication. It would be more readable.

We are really sorry for any misunderstanding, the 16 studies mentioned are in vitro on human cells and not in vivo: of the 33 included studies, 5 were in vivo and 28 in vitro, and of the latter, only 16 used human cell populations;

We corrected the sentence as recommended, as follows: “Of the in vitro studies, the following 16 papers on the effects of zinc were on human cell populations: gastro-intestinal apparatus using liver cancer cell line [49], in the orofacial district were analyzed gingival fibroblasts [36, 40, 53-56], oral keratinocyte cells [44] … “

Table 3

It will be good to add

Doneria  et al; 2017

We corrected the Table 3 and 4 as recommended.

Line 216

MIC and MBC- please explain these aberrations

sulfide (H2S- sulfide (H2S

We are sorry for the misunderstanding, MIC is minimum inhibitory concentrations (explain for the first time at line 213) and MBC is minimum bactericidal concentration (explain at line 214) and we corrected all the wrong chemical formula of H2S in the manuscript as pointed out.

I would divide the3.2.1. Conservative and cariology  paragraph into even smaller paragraphs: e.g. odor, cytotoxicity, physical properties of materials after the addition of Zn, etc., it would be more readable.

  I very often refer to review works and read only those paragraphs that I am most interested in, to make it easier to find source publications. This precise division of information into subheadings makes reading very easy.

We thank the reviewer for the suggestions and divide the paragraph in three sections.

Line 254

Ag revealed- in previous sentence you used Ag, now you can write silver .

  1. faecalis- E. faecalis

Zn2+,- Zn2+. Garcia et al

We thank the reviewer to have pointed out these mistakes and corrected them.

Discussion

Line 361

especially when used in pedodontics, and it would be crucial to combine a good antibacterial-  or periodontics?- The entire article concerns dentistry, so the conclusions will not include the effect of zinc on feet?

it is crucial that materials have better mechanical properties also containing an antibacterial because isolation during paediatric procedures is essential but not always achievable with young, special needs or uncooperative patients.

Line 374

have evaluated the use of zinc as a coating material for periodontal or implant-prosthetic

use.  Costing material for implant I agree, but for the surface of the root this is the new info for me, please add ref to it.

We are really sorry, there is a mistake, we have eliminated the wrong word “periodontal” added in the sentence.

Line 383

  1. mutans- S. mutans

We made the recommended correction.

This manuscript is a resubmission of an earlier submission. The following is a list of the peer review reports and author responses from that submission.

Round 1

Reviewer 1 Report

Comments and Suggestions for Authors

I have no additional comments.

I have found one small mistke, line 280 should be d L. acidophilus

Reviewer 2 Report

Comments and Suggestions for Authors

·      I’ve reviewed the paper titled "Systematic review of zinc benefits and biological effects on oral health" that provides a comprehensive review of the benefits and biological effects of zinc on oral health. Please find my suggestions and some comments on this manuscript:

·      The authors have analyzed zinc application in dental materials, with different compositions and chemical formulations, considering how mechanical and biological properties may influence its clinical applicability. The study concludes that antibacterial activity was found to be the most studied property of zinc, but further investigation is needed to establish adjuvant zinc therapies in patients with oral disease. As a reviewer, I would suggest that the authors should provide more information on the potential toxicity of Zn nanomaterials and review the in vivo studies to confirm the properties of zinc-containing dental materials.

·      The manuscript is well written but I would suggest that the authors should formulate a focused question based on the PICOS strategy and explain how it relates to the current literature and practice.

·      The authors should use tables and figures to summarize and present the data in a clear and concise way.

·      The tables in the manuscript are very informative and detailed. However, Table 1 does not provide the full PICOS question and the rationale for each element. Table 2 and 3 can be improved further by including the sample size, the intervention and control groups, and the outcomes of each study. Table 4 may also include the cell types or microbial strains, the specimen characteristics, and the control group of each study.

·      The discussion section highlights the main findings and implications of the study, as well as the limitations and suggestions for future research. However, the authors are encouraged to compare their results with other systematic reviews or meta-analyses on the same topic, and mention any conflicting or contradictory evidence related to the applications of application of zinc in several dental application discussed in this manuscript.  

·      The conclusion section is concise and effective.

·      The authors use appropriate citations and references.

Reviewer 3 Report

Comments and Suggestions for Authors

Dear editors,

MY points are still the same about this article:

The article lacks more details about search strategy, selection criteria, inclusion criteria, exclusion criteria, data extraction method, risk of bias and data synthesis.

·         Why is there no meta-analysis?

·         Why have you used PubMed as a unique database?

Comments on the Quality of English Language

none

Reviewer 4 Report

Comments and Suggestions for Authors

Dear authors, thank you for all the answers. Based on the new version presented, I also present considerations:

The search strategy was applied in three different scientific databases, as mentioned. However, the entire text needs to be updated according to these new searches. For example: Table 2 – only shows the search strategy applied in PubMed; figure 1 – outdated.

“The search strategy presented does not include the types of studies defined in the PICOS strategy.” “We rechecked that the types of items excluded from the strategy search were in line with the established PICOS.”

The authors claim that they checked the search strategy with keywords related to study design excluded, however, I emphasize the need to have included the types of studies you wish to include in the systematic review. According to the figure 1, 11 studies were manually inserted. This quantity reflects the need to improve the search strategy.

“We did not apply the search only to the PubMed search engine, but on Scopus and Web of Science there were very few results and still included in PubMed.” Dear authors, the construction of a systematic review requires a wide of scientific searches. In addition, even when few studies are observed on certain scientific bases, such bases should not be disqualified and not included. Furthermore, the removal of duplicate studies should be considered in the methodology of any systematic review.